# Comparison of 0.12% Chlorhexidine and a New Bone Bioactive Liquid, BBL, in Mouthwash for Oral Wound Healing: A Randomized, Double Blind Clinical Human Trial

**DOI:** 10.3390/jpm12101725

**Published:** 2022-10-16

**Authors:** Eduard Ferrés-Amat, Ashraf Al Madhoun, Elvira Ferrés-Amat, Neus Carrió, Miguel Barajas, Areej Said Al-Madhoun, Eduard Ferrés-Padró, Carles Marti, Maher Atari

**Affiliations:** 1Oral and Maxillofacial Surgery Service, Hospital HM Nens, HM Hospitales, 08009 Barcelona, Spain; 2Oral and Maxillofacial Surgery Department, Universitat Internacional de Catalunya, St Josep Trueta s/n, Sant Cugat del Vallès, 08195 Barcelona, Spain; 3Department of Animal and Imaging Core Facilities, Dasman Diabetes Institute, Dasman 15462, Kuwait; 4Department of Periodontology, Universitat Internacional de Catalunya (UIC), C/Josep Trueta s/n, Sant Cugat del Valles, 08195 Barcelona, Spain; 5Biochemistry and Molecular Biology Department, Universidad Pública de Navarra, 31006 Pamplona, Spain; 6Biointelligent Technology Systems SL, C/Diputaccion 316, 3D, 08009 Barcelona, Spain; 7Oral and Maxillofacial Surgery Department, Hospital Clinic de Barcelona, 08036 Barcelona, Spain; 8Invesbiofarm, PTS Parque Tecnológico, Armilla, 18016 Granada, Spain; 9Ziacom Medical SL, C. Buhos, 2, 28320 Madrid, Spain

**Keywords:** chlorhexidine, bone bioactive liquid, BBL, Theravex, oral wound healing, pain index score, early wound healing index score

## Abstract

Following surgery, healing within the oral cavity occurs in a hostile environment, and proper oral care and hygiene are required to accelerate recovery. The aim of the current study is to investigate and compare the bioreactivity characteristics of mouthwashes based on either chlorhexidine (CHX) or a novel bone bioactive liquid (BBL) in terms of oral healing within seven days application post-surgery. A randomized, double blind clinical trial was conducted in 81 patients, wherein the mouthwashes were applied twice a day for a period of 7 days. The visual analog scale (VAS) protocol was applied to determine pain index scores. Early wound healing index (EHI) score was determined for evaluating oral cavity healing progress. No adverse effects were observed using the mouthwashes, but CHX application resulted in stained teeth. Applications of both CHX and BBL were sufficient to reduce pain over a period of 7 days. However, the BBL group demonstrated a statistically significant reduction in VAS scores starting on day 4. The EHI scores were significantly higher in the BBL group compared with the CHX group, independent of tooth location. No differences in either VAS or EHI scores due to gender were observed. Compared with the commercially available CHX mouthwash, application of the BBL mouthwash reduced pain and accelerated oral cavity healing to a greater extent, suggesting it effectively improves the oral cavity microenvironment at the wound site in mediating soft tissue regeneration.

## 1. Introduction

Wound healing requires a chronological sequence of complex biological processes [1]. All tissues in the body essentially follow the same pattern to promote healing, with a focus on quick recovery [2]. Nevertheless, these processes are dependent on intact hemostatic and inflammatory mechanisms that are widely influenced by genetic and environmental factors, especially in cases of wound healing that concern the oral cavity, which is characterized by a remarkably hostile environment based on its resident microbiome [3]. 

Following oral surgery or tooth extraction, a sequence of healing processes are immediately initiated. The periodontal pocket is blocked by blood coagulation [4], and a re-epithelization mechanism is initiated, followed by granulation tissue generation [5]. After one week of tissue remodeling, bones replenishment occurs, and cavity closure is completed within a period of eight weeks after tooth extraction [6,7]. 

Notably, several factors interfere with healthy oral healing processes, including the tooth location, smoking, and mouth care attitudes [8,9]. Therefore, effective oral care and hygiene are crucial after surgery to minimize pain, inflammation, and dental plaque formation [10]. Nevertheless, it can be a challenge for patients to maintain sufficiently high hygiene [11]. Thus, efficient would healing detergents are necessary to sustain and accelerate recovery after oral surgery.

Nonprescription dental hygiene products are available and are normally sufficient for preventing common oral health problems. Chlorhexidine (CHX) is the most common antiplaque and antigingivitis agent [12]. CHX is a cationic bisbiguanide compound with broad-spectrum antimicrobial properties. It binds to the microbe cell and precipitates the cell contents [13,14,15]. CHX-gluconate is widely used in dentistry and is available as an oral rinse, gel, spray, and dental varnish. In their recent review article, Rajendiran et al. have summarized the current developments in antiplaque, antigingivitis, and antiperiodontitis properties of CHX and other compounds [16].

Bone bioactive liquid (BBL) is a saline solution containing calcium chloride (CaCl_2_) and magnesium dichloride hexahydrate (MgCl_2_-6H_2_O) with a net negative charge that promotes healing and soft and hard tissue regeneration in the wounded periodontal cavity [17]. Furthermore, BBL significantly intensifies the concentration of hydroxyl groups at the wound surface and significantly improves hydration in comparison with other mouthwashes (unpublished data). BBL creates a hydrophilic environment that allows active ionic interactions with blood plasma, progenitor endothelial, and epithelial cells and, consequently, the coordination and communication between cells are significantly improved at the wound site [17].

The aim of the present study is to compare the efficacy of BBL and CHX (0.12%) mouthwashes in improving clinical parameters and soft tissue healing after tooth extraction. The wound healing properties of BBL may support its usage as a new pharmaceutical product with good physical, chemical, and biological stabilities.

## 2. Materials and Methods

### 2.1. Study Population: Inclusion and Exclusion Criteria

The study cohort comprised 81 patients of male and female genders aged above 14 years old who had agreed to voluntarily participate in the clinical trial. The sociodemographic characteristics of the patient cohort are described in Appendix A. Written informed consent was obtained from all study participants following the ethical guidelines of the Declaration of Helsinki and approved by the ethics committee at Complejo Hospitalario de Toledo and institutional review board, Spain (CEIm HM Hospitales 21.03.1786-GHM; protocol ID: V01-2021; date: 16 April 2021; Clinical Trial Registry Platform: Clinical Trial Gov. Press). Participant inclusion criteria included the following: systemically healthy, full mouth plaque and bleeding scores < 20%, healthy periodontium, and no local or systemic antibiotic or antiseptic treatments for 3 months prior to involvement in the study. Exclusion criteria included the use of medications that cause gingival enlargement or the presence of gingival idiopathic overgrowth; smokers; patients with systemic diseases or conditions that could interfere with routine periodontal therapy such as pregnancy or lactating females, uncontrolled periodontal disease, previous or current history of bisphosphonate treatment, immune deficiencies, uncontrolled diabetes, rheumatoid disease, radiotherapy, chemotherapy, infectious diseases. 

### 2.2. Removal of Patients from Therapy or Assessment

Participants were free to withdraw from the study at any time, without any prejudice or justifications. If the patient prematurely discontinued the study, any relevant evaluations and observations and reasons for study discontinuation were recorded in the case report form (CRF). Participants discontinuing due to infection or medical reasons were monitored until complete recovery.

### 2.3. Study Design

The study was designed as a one-week randomized, prospective, double blind pilot clinical trial. This prospective study included patients who required two trans alveolar surgical extractions of inferior or superior third molar or any simple or surgical tooth extraction. The 171 dental extractions in 81 patients were randomly assigned to two groups: the control group (CG, 20 male and 22 female patients) received Perio-Aid Intensive Care mouthwash containing 0.12% CHX-di-gluconate (Dentaid, Barcelona, Spain); and the test group (TG, 19 male and 20 female patients) received BBL mouthwash, a bioactive solution generated in our laboratory, which is phosphate-buffered saline (PBS) solution containing 1.35 mM CaCl_2_ and 0.75 mM MgCl_2_-6H_2_O with a net negative charge. Mouthwashes were administered twice a day for 7 days, and no eating or drinking was permitted for a period of 1 h after the treatment. After a period of 7 days, clinical parameter data were analyzed to determine clinical changes during the treatment. Patient follow-ups were conducted twice via phone calls at days 2 and 4 to determine the degree of postoperative pain as described in Table 1. To respect patient data confidentiality, the data management system described in [18] was applied.

### 2.4. Surgery Assessments

Three independent dentists participated in clinical data measurements and registrations. The study participants received a diagnostic workup including clinical examinations, oral photographs, and standardized periapical radiographs to evaluate the proposed surgical sites. Before the surgical procedure, patients underwent periodontal therapy and received extensive oral hygiene instructions for providing an improved oral environment. The protocols for full mouth plaque scores (FMPS) and full mouth bleeding scores (FMBS) were implemented exactly as described by T.J. O’Leary et al. [19] and J. Ainamo et al. [20], respectively, and were recorded after the hygienic phase of the periodontal therapy. No surgery was performed until patients reached FMPS < 20% and FMBS < 20%. Each patient received surgery on either of the two bilateral areas on different days. The surgical extractions of tooth were carried out with local anesthetic, raising a mucoperiosteal flap with osteotomy, and no periodontal dressing was applied postoperatively. Unless otherwise required, dental extractions were performed without stitches in both control and test groups to evaluate the healing capacity of both mouthwashes.

### 2.5. Post-Surgical Procedures

All the patients received 600 mg ibuprofen every 8 h for 4 days and either 500 mg amoxicillin every 8 h for 7 days or 100 mg doxycycline every 24 h for 5 days for patients allergic to amoxicillin. Patients were instructed to rinse with 15 mL mouthwash twice a day after their regular homecare practice for a period of 7 days. The use of ice packs was recommended for at least 3 h post-surgery. All the patients were instructed to discontinue tooth brushing at the surgical sites for 7 days. After 30 days, a professional prophylaxis was performed to remove stains caused by Perio-Aid Intensive Care mouthwash.

### 2.6. Wound Healing Measurement Procedures

After a period of 7 days, patients were examined for evaluation of healing. The early wound healing index (EHI) [21] scores were determined by two blinded clinical examiners. The scale was applied with five different degrees, and scores 5 to 1 were applied based on the respective observations: complete flap closure without fibrin line; complete flap closure with fibrin line; complete flap closure with small fibrin clot(s); incomplete flap closure with partial necrosis; and incomplete flap closure with complete necrosis (more than 50% of the former flap is involved). In addition, EHI was assessed using the healing index of Landry et al. [22], in which wounds were graded on a scale of 1–5 as described in Appendix A. The wound area was classified as either partially or fully keratinized. In the case of partial keratinization, the wound area was further classified as partially or fully keratinized upon examination after an additional 7 days. 

### 2.7. Procedures for Measurement of Post-Surgical Pain, Safety, and Discomfort 

Efficacy measurements were assessed by pain scale evaluation post-surgery at days 2, 4, and 7 though a phone call with the patients in accordance with their subjective pain feeling. A modified visual analog scale (VAS) was applied as described in [23]. No pain was scored as 0, moderate pain scale as 5, and maximum pain as 10. Furthermore, safety measurements were evaluated by the incidence of adverse events (AEs) and serious adverse events (SAEs) that could be detected by the investigator or communicated by the patient throughout the entire study.

### 2.8. Statistical and Analytical Methods 

Shapiro–Wilk normality testing was performed to assess the normality of data distribution. Data are reported as mean and standard deviation or median and interquartile range (IQR) based on data distribution. Differences between treatments and gender were assessed using Mann–Whitney test or t-test based on data distribution. A two-tailed test with a *p*-value < 0.05 was considered the cut-off level for indicating statistical significance. Statistical Package for Social Sciences software (IBM SPSS, version 23, Chicago, IL, USA) was used for data analysis.

## 3. Results

In this study, no incidences of adverse events were observed, and no postoperative compilations were reported. Patients did not present statistically significant differences in terms of infection prevention between the CG and TG (*p* = 0.96).

### 3.1. BBL Mouthwash Dramatically Reduces VAS 

In general, patients from both treatment groups showed a progressive decrease in pain over the consecutive week post-surgery (Figure 1A). Nevertheless, statistical data analysis revealed that the CHX group showed a significant reduction in VAS only at day 7 (*p* < 1 × 10^−5^). Alternatively, the BBL group demonstrated a significant reduction in VAS starting on day 4 (*p* < 1 × 10^−4^) and was further reduced at day 7 (*p* < 1 × 10^−8^, Figure 1A).

VAS at day 2 was lower in the BBL group than in the CHX group, though not significantly (*p* = 0.084). However, significant differences in VAS scores were observed at days 4 and 7, with notable 50–70% lower values scored in the BBL group compared with the CHX group (Figure 1A).

Since the study compromises patients from two different genders, we were interested to evaluate the VAS in each separately. VAS was comparable between males and females and no significant differences were recorded (*p* = 0.78). The time course study indicated that the VAS had an identical trend in both genders, with a significant reduction starting at day 4 for the BBL group and starting at day 7 for the CHX group (Figure 1B,C).

### 3.2. BBL Mouthwash Improves EHI

The total extracted teeth were 89 and 82 for the BBL and CHX groups, respectively. As observed in Figure 2, both CHX and BBL treatments improved oral wound healing at day 7 post-surgery, independent of the number of extracted teeth. Nevertheless, wound closure was notably enhanced in response to BBL treatment. Table 2 shows the EHI scores based on the Landry et al. [22] classification for a total of 171 dental extracts performed in the patients cohort. In general, and independent of the position of dental extraction, the EHI scores were remarkably higher after BBL treatment than after CHX treatment. Taken together, these results indicate that BBL treatment enhances gingival tissue healing.

We then evaluated the EHI scores for both treatments. Data analysis revealed a statistically significant differences in EHI scores between the two treatments. The average of the total EHI score was 4.40 ± 0.56 for the BBL group and 3.1 ± 0.57 for the CHX group, indicating the remarkable healing process occurring in the BBL group.

Since the tooth location influences the EHI score, we classified the extracted teeth into premolars, centrals, and molars and accordingly determined the EHI scores. As shown in Figure 3A, the EHI score for each tooth segment was significantly higher in the BBL group compared with the CHX group. In the comparisons between tooth segments, molars had the lowest EHI scores, indicating a delayed healing process relative to the other tooth locations (Figure 3A).

No significant differences in EHI scores were detected between genders (*p* = 0.645). Both males and females showed identical patterns with respect to the tooth location segments, where a significant improvement in EHI score was observed in the BBL group relative to that of the CHX group (Figure 3B,C).

## 4. Discussion

In our previous preclinical study, we applied pretreated bone level tapered (BLT) titanium implants in foxhound dogs with BBL. The data indicated that BBL improves the histological and histomorphometric characteristics of the implants, reduces titanium surface roughness, improves wettability, and promotes healing and soft and hard tissue regeneration at the implant site [17]. In the current study, BBL was applied as a mouthwash to human patients who had teeth extraction surgeries, and its prospective clinical properties were compared to that of CHX mouthwash. Overall, the human data support the previous findings of the animal study and indicate significant improvements in wound healing and soft tissue regeneration from BBL. Notably, the study patient cohort consisted of a heterogenous population with different sociodemographic backgrounds that were randomly distributed in this double blinded study.

CHX is a gold standard mouthwash with antiplaque and antigingivitis properties [24,25]. Nevertheless, it has negative side effects that preclude long-term use and result in poor patient compliance [25]. The adverse effects of CHX include burning sensation, tartar and calculus formation, soft tissue trauma and allergy, taste alteration, and teeth staining [26,27,28,29]. Therefore, studies have been directed toward the use of different of CHX concentrations and/or alternative product usage. Several studies compared the use of different CHX concentrations, which are thoroughly reviewed in [29,30,31]. In general, studies indicate that there are no statistically significant differences in the efficacy of 0.12% and 0.2% CHX mouthwashes, and that concentrations above 0.2% will unnecessarily increase the incidence of unwanted side effects. Alternatives to CHX include cetylpyridinium chloride, oxidizing mouthwashes, and povidone-iodine (PVP-I). Although, some of these products show less adverse effects, their application is limited by the absence of clinical research, such as randomized clinical trials and systematic meta-analysis reviews, or the absence of commercially available formulations for intraoral use. 

Bioactive BBL is a new commercial mouthwash solution that is basically a negatively charged liquid saline containing Ca^2+^ and Mg^2+^ salts. BBL has no taste, no odor, and does not induce allergic reactions. Comparison of the efficacy of CHX and BBL in human patients revealed that the latter dramatically reduces pain within a period of 4 days and promotes complete oral healing with 7 days. The current randomized, double blind clinical trial indicates that BBL is an effective mouthwash solution for prospective clinical applications. Nevertheless, further longitudinal studies are required to delineate its capacity based on antiplaque and antiseptic properties.

## 5. Conclusions

Clinical data collected from patient diaries revealed a statistically significant positive effect for the BBL mouthwash in improving post-operative quality of wound healing compared with CHX mouthwash both during and after 7 days of application. The application of both mouthwashes resulted in differential progressive pain reduction in consecutive weeks post-surgery; however, BBL resulted in significant pain relief starting at day 4. No gender differences associated with pain or would healing were observed in response to the applied mouthwashes. Together, the wound healing properties of BBL may support its usage as a new pharmaceutical product with good physical, chemical, and biological stabilities.

## 6. Study Limitation

On the follow up evaluation, patient assessments were conducted based on phone calls, which may raise some concerns about bias related to early health status post-surgery. Although, the clinical observations were performed by several specialized dentist who were blinded for the treatment type, bias concerns may also be raised about the VAS analysis. In addition, as part of general clinical practice, all patients were treated with antibiotics in addition to the study treatment, which may suggest that the observed study outcome is due to an effect of both the treatment and the antibiotic.

## Figures and Tables

**Figure 1 jpm-12-01725-f001:**
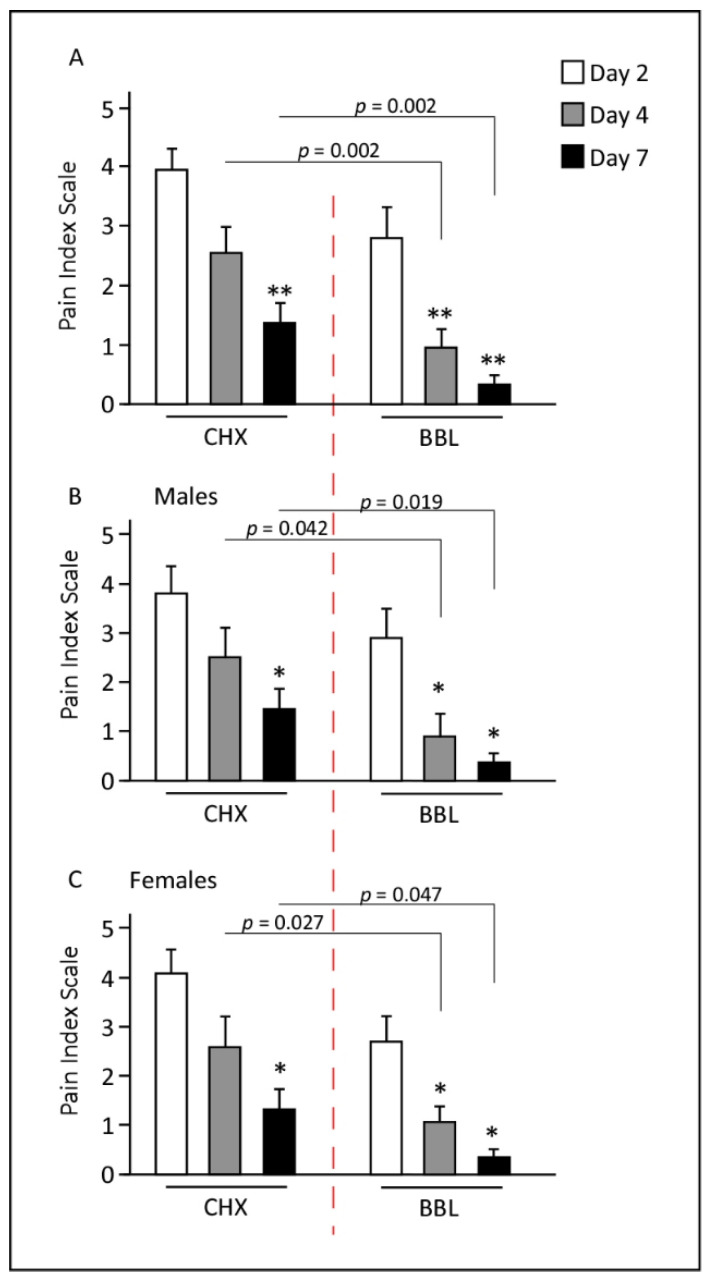
Pain visual analog scale (VAS) scores were reposted at days 2, 4, and 7 from the day of surgery. (**A**) Both CHX and BBL groups showed significant improvements in pain scores, though the BBL group VAS scores were significantly improved relative to the CHX group. (**B**,**C**) Study comparing between genders. Males and females responded similarly for both treatments, and the VAS response trend was comparable. * *p* < 0.05; ** *p* < 0.002.

**Figure 2 jpm-12-01725-f002:**
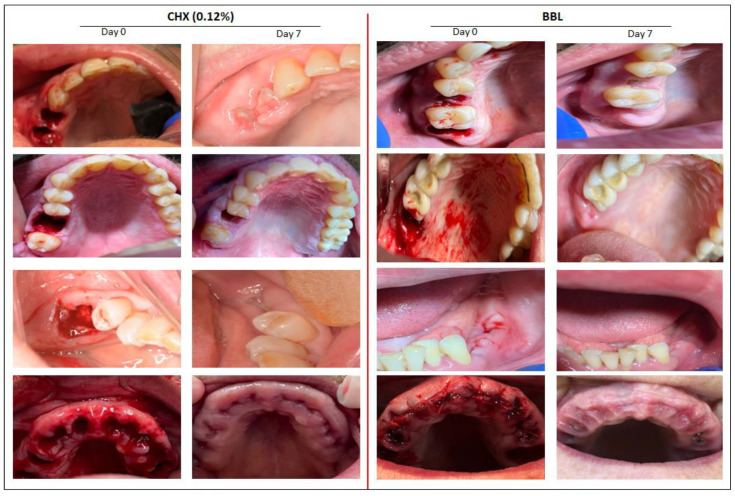
Representative images for patients at the surgery day 0 and after 7 days using CHX or BBL mouthwash. Notable wound healing improvements were detected in patients who used BBL mouthwash.

**Figure 3 jpm-12-01725-f003:**
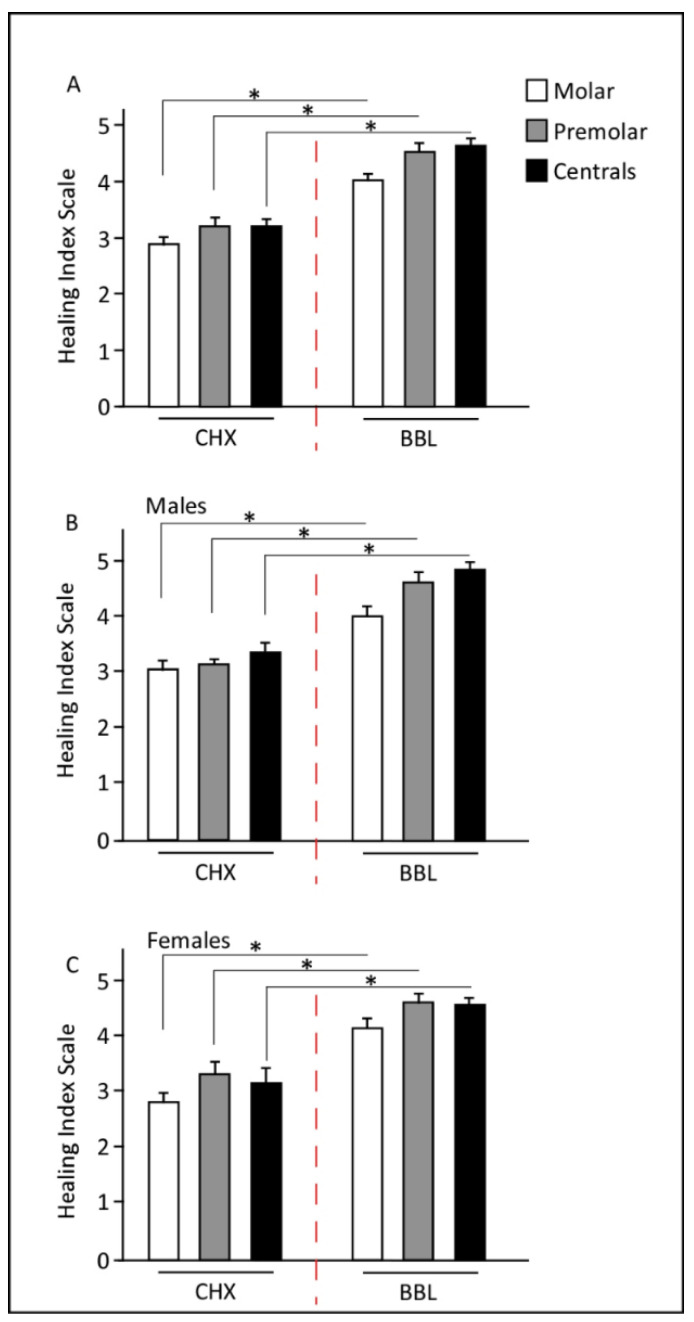
Early wound healing index (EHI) scores at day 7. (**A**) EHI scores were independent of tooth position and were significantly higher in BBL treatments. (**B,C**) Comparison between genders. Males and females responded similarly for both treatments, with a significant improvement in EHI post BBL treatment. * *p* < 1 × 10^−4^.

**Table 1 jpm-12-01725-t001:** Study design.

Criteria	Visit 1	Phone Calls		Visit 2
	(Surgery Day)	Day 2	Day 4	Day 7
Informed consent	X			
Inclusion/exclusion criteria	X			
Collection of clinical data	X			X
Oral examination	X			X
Patient diary: Pain VAS scale 0–10		X	X	X
Patient diary: clinical healing measurements	X			X
Tolerability and post-treatment side effects				X

**Table 2 jpm-12-01725-t002:** The early wound healing index (EHI) scores were determined as described by Landry et al. 1988.

EHI Score	Total of 171 Dental Extractions in Operations on 81 Patients	
	BBL Treatment	0.12% Chlorhexidine Treatment
1—very poor	0	0
2—poor	0	4
3—good	5	64
4—very good	49	9
5—excellent	38	2

## Data Availability

The datasets used and/or analyzed during the current study are available from the corresponding author on reasonable request.

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
