# Peer review of "Comparison of 0.12% Chlorhexidine and a New Bone Bioactive Liquid, BBL, in Mouthwash for Oral Wound Healing: A Randomized, Double Blind Clinical Human Trial"

_jpm, 2022, doi:10.3390/jpm12101725_

Round 1
Reviewer 1 Report
Authors conducted a clinical trial to contrast the effects between chlorhexidine (CHX) and a bone bioactive liquid (BBL) on wound healing and pain, after dental extraction surgery. Pain was evaluated by using a modified visual analog scale (VAS), while wound healing was assessed with an early wound healing index (EHI). Compared to CHX, authors found that BBL start reducing pain 3 days before. Also, authors found that BBL significantly increase wound healing after one week of treatment, compared to CHX.
-English writing is poor. Grammar check on the entire manuscript should be performed.
-Concentration of calcium chloride and magnesium chloride in BBL should be provided.
-Authors mention that “...wound closure and tissue inflammation were notably enhanced in response to BBL treatment...” (lines 212-213). How did authors measure inflammation and why did BBL treatment increase inflammation?
-Sociodemographic characteristics of the patients should be reported in a supplementary table
-Scores of the EHI should be provided in detail. I suggest to show a table with each score from 1 to 5, indicating whether granulation tissue is present, % of red to pink gingiva, epithelialization of the incision margin vs exposed connective tissue, for instance.
-In figures 3B and 3C, the Y axis should end at 5.
Author Response
Reviewer 1
Comments and Suggestions for Author
Authors conducted a clinical trial to contrast the effects between chlorhexidine (CHX) and a bone bioactive liquid (BBL) on wound healing and pain, after dental extraction surgery. Pain was evaluated by using a modified visual analog scale (VAS), while wound healing was assessed with an early wound healing index (EHI). Compared to CHX, authors found that BBL start reducing pain 3 days before. Also, authors found that BBL significantly increase wound healing after one week of treatment, compared to CHX.
- -English writing is poor. Grammar check on the entire manuscript should be performed.
We understand the reviewers concerns to improve our manuscript; therefore, our plan is to submit the final version of manuscript to the English language editing facility provided by MDPI service.
- -Concentration of calcium chloride and magnesium chloride in BBL should be provided.
We thank the reviewer for his/her noticed. BBL contains 1.35 mM CaCl2, and 0.75 mM MgCl2-6H2O. The concentrations were added at Page 3-4, lines 117-118.
- -Authors mention that “...wound closure and tissue inflammation were notably enhanced in response to BBL treatment...” (lines 212-213). How did authors measure inflammation and why did BBL treatment increase inflammation?
We thank the reviewer for the noticed. BBL treatment resulted lower gingival inflammation as noticed in Figure 1. Typo error. Since we did not study inflammatory markers, we decided to rephrase the sentence.
- -Sociodemographic characteristics of the patients should be reported in a supplementary table.
Excellent notice by the reviewer. Yes, indeed the Sociodemographic characteristics are important in our study. Unfortunately, we forget to submit in the 1st version. Please, see the supplemental Table S1, Material and Methods section (Page 3, lines 86-87), and Discussion section (Page 9, lines 259-261).
- -Scores of the EHI should be provided in detail. I suggest to show a table with each score from 1 to 5, indicating whether granulation tissue is present, % of red to pink gingiva, epithelialization of the incision margin vs exposed connective tissue, for instance.
Excellent notice by the reviewer. The EHI scores were in accordance with Landry et. al. (1985) classifications. Please see Supplemental T2. We also added Table 2 (Page 7) and modified the Material and Methods section (Page 5, Lines 161-162), Results section (Page 7, Lines 217-221)
- -In figures 3B and 3C, the Y axis should end at 5.
We thank the reviewer for the notice. We corrected the Figure as requested (Page 8).

Reviewer 2 Report
Thank you for submitting your manuscript and your efforts, it is an interesting study with good info on mouth rinses
The references need to be in Vancouver style
Spell check your work throughout the manuscript
Good basis of literature set up to get to the aims of study
Well explained process of methodology and study design, however, phone call to follow up patients’ data collection can be biased? Did you address this?
The antibiotic coverage might camouflage the results obtained from mouth rinses? Please address this.
Please introduce the full word before the use of abbreviation
The lit review on dog study makes sense but this clinical study based upon VAS might include biasness and more stronger evidence based analytical techniques might prove the efficacy of mouthwash over chlorhexidine which needs to be clearly mentioned
Please add significance of the study at the end of discussion.
Author Response
Comments and Suggestions for Authors
Thank you for submitting your manuscript and your efforts, it is an interesting study with good info on mouth rinses
- The references need to be in Vancouver style
We thank the reviewer for the notice. We corrected the reference style as requested.
- Spell check your work throughout the manuscript
We understand the reviewers concerns to improve our manuscript; therefore, our plan is to submit the final version of manuscript to the English language editing facility provided by MDPI service.
- Good basis of literature set up to get to the aims of study
We thank the reviewer for the complement.
- Well explained process of methodology and study design, however, phone call to follow up patients’ data collection can be biased? Did you address this?
We thank the reviewer for his/her concerns. We agree with the reviewer phone call evaluations may create biased evaluation specially at earlier stages post-surgery. In this revised version of the MS, we referred this issue as study limitation (Section 6), Please see, Page 10, lines 296-301. Nevertheless, the in-clinic visit at day 7, confirmed the welfare of patients’ health status.
- The antibiotic coverage might camouflage the results obtained from mouth rinses? Please address this.
We thank the reviewer for his/her concerns. Administration of antibiotic is a common practice in this type of surgeries. Therefore, the observed effect is due to both the treatment and the antibiotic usage. Therefore, in this revised version of the MS, we referred this issue as study limitation (Section 6), Please see, Page 10, lines 296-301.
- Please introduce the full word before the use of abbreviation
We thank the reviewer for the notice. We corrected the missing full words.
- The lit review on dog study makes sense but this clinical study based upon VAS might include biasness and more stronger evidence based analytical techniques might prove the efficacy of mouthwash over chlorhexidine which needs to be clearly mentioned.
We thank the reviewer for the noticed and we understand the reviewers concerns to improve our manuscript. In this human clinical study, the VAS was performed by several specialized dentists, who were blinded for the treatment type, also the statistical analyses were done by other scientists to avoid possible conflicts (Included in the Study limitation section Page 10, lines 296-303).
We would be happy if the reviewer would suggest some analytical techniques to perform in our future studies. Currently, we are conducting a multi-centers clinical study in several countries with a much larger participants (N), in addition to VAS, we are collecting samples (swap) from the treated areas on day 3 and day 7 post-surgery to compare the levels of inflammatory factors and micro-organisms after BBL and 12% chlorhexidine mouth rinse. We currently do not have yet sufficient data to be included in this brief report.
- Please add significance of the study at the end of discussion.
We thank the reviewer for his/her noticed, we add the significance of the study at the discussion section (Page 9, lines 281-284) and conclusion section (Page 10, lines 293-294).

Round 2
Reviewer 2 Report
Thanks for modifications, the title of the article is still lengthy, may be just remove the word BBL.
Study limitation-may be start the sentence with on follow up evaluation instead of day 2 or 4 as when you end the article, it has to be more professional writing.
Author Response
Dear Reviewer:
We appreciate your kind support to imporve our manuscript.
- Thanks for modifications, the title of the article is still lengthy, may be just remove the word BBL.
We added the BBL to the title as a keyword that will help future citation and article referal. We undestand that the title is long, but it reflecats the overall of the study. We would appraite any suggestion that may imporve it.
2. Study limitation-may be start the sentence with on follow up evaluation instead of day 2 or 4 as when you end the article, it has to be more professional writing.
We agree with the Reviewer. Pleaase, find the new English language edited manuscript by the MDPI editing survice. We corrected the suggested concern. Please see Page 11, line 317.
Best REgards,
Ashraf